# Location Transparency Call (LTC) System: An Intelligent Phone Dialing System Based on the Phone of Things (PoT) Architecture

Haytham Khalil * and Khalid Elgazzar *

Faculty of Engineering and Applied Science, Ontario Tech University, Oshawa, ON L1G 0C5, Canada
* Correspondence: haytham.aboulabbas@ontariotechu.net (H.K.); khalid.elgazzar@ontariotechu.ca (K.E.)

**Abstract:** Phone of Things (PoT) extends the connectivity options for IoT systems by leveraging the ubiquitous phone network infrastructure, making it part of the IoT architecture. PoT enriches the connectivity options of IoT while promoting its affordability, accessibility, security, and scalability. PoT enables incentive IoT applications that can result in more innovative homes, office environments, and telephony solutions. This paper presents the Location Transparency Call (LTC) system, an intelligent phone dialing system for businesses based on the PoT architecture. The LTC system intelligently mitigates the impact of missed calls on companies and provides high availability and dynamic reachability to employees within the premises. LTC automatically forwards calls to the intended employees to the closest phone extensions at their current locations. Location transparency is achieved by actively maintaining and dynamically updating a real-time database that maps the persons' locations using the RFID tags they carry. We demonstrate the system's feasibility and usability and evaluate its performance through a fully-fledged prototype representing its hardware and software components that can be applied in real situations at large scale.

**Keywords:** Internet of Things (IoT); Phone of Things (PoT); Asterisk; chatbot; RFID; Message Queuing Telemetry Transport (MQTT); MongoDB

## 1. Introduction

Finding the right employee at the right time is a crucial challenge for almost every business. The time spent locating a specific employee within the premise can lead to expensive downtime, loss of potential revenue, missed deadlines, slower workflow, less customer satisfaction, and lowering of reputation. Statistics reveal that 80% of all business transactions are carried out over the phone [1]. According to [2], a missed call costs £1200 in trade loss. They estimate that UK businesses, for example, lose over £30 billion each year as a direct effect of missed business calls. Likely, the primary cause of unanswered business phone calls is the lack of availability of the right person to answer the call or the lack of service capacity to digest a large number of incoming calls [3]. The latter is simple, and it requires consultation with the service provider to upgrade the system capacity. The former, however, requires out-of-the-box solutions to map customer inquiries to the right staff intelligently. Intelligent forwarding of customer inquiries to the right employees overcomes the notorious bottleneck queuing functionality that businesses usually implement using the automated-attendant feature [4] embedded within the unified communication (UC) system within the premises. According to [5], 75% of customers were exceedingly irritated and hung up when they could not get somebody on the phone in a sensible amount of time. Moreover, 85% of the people whose calls are missed will not call back. For businesses, this means that any potential revenue that could have been received from that missed call is now lost, which quickly adds up to a significant amount.

The Interactive Voice Response (IVR) [6] system has been extensively used by businesses to optimize call queuing. However, IVR uses a static configuration that needs to be

rebuilt every time a change is required. Moreover, customers usually report frustration regarding option tree changes, confusion, routing calls to wrong places, or long waiting times after being routed to the intended representative.

Third-party platforms, such as Twilio [7], have been utilized to provide phone interface capabilities to businesses and applications. These platforms provide APIs to build interactions with customers on the preferred channels. However, third-party platforms lack the flexibility needed to adapt to the dynamic requirements of applications. Nevertheless, they incur an extra cost that hinders their widespread usage. PoT overcomes the drawbacks of third-party services and provides an interface to the existing phone network infrastructure within the premises of enterprises at no additional cost. PoT enables adaptive phone integration that fits the needs of different applications. Therefore, rather than relying customer requests to a static inbound phone number of the business, which is the case with third-party platforms, PoT can be programmatically tuned to route calls to arbitrarily existing extensions within premises based on predefined criteria.

This paper presents the Location Transparency Call (LTC) system. LTC is an application of Phone of Things (PoT) [8]. LTC provides an intelligent solution for companies to timely reach their employees regardless of their mobility within the enterprise's premises. The intended employees are reached by simply invoking their names or the customer inquiries they can handle to the embedded chatbot agent and thereby to the PoT gateway. If no name is mentioned, the system will try its best to match based on the required job functionality and employees availability. The LTC system then automatically locates the intended employees within the premises and forwards the calls to the closest extension numbers to their current locations. To accomplish this, the system actively keeps track of all employees while they move within the premises and dynamically updates a real-time database that maps employees to the closest extension number(s).

Since the emergence of telephones, people have been identified by their unique extension numbers. This requires the remembrance of the extension numbers of the called persons, which is daunting, and in many cases, infeasible. Nevertheless, the employees will likely miss calls if they leave their default locations. UCs provide find-me/follow-me features to mitigate missed calls and provide relatively high availability to called persons. The find-me feature routes incoming calls to a static pre-configured extension. However, the follow-me feature distributes incoming calls to a set of statically pre-configured extensions at once. In either case, however, these features still lack adaptability to an individual's mobility; since they are static configurations that need to be manually updated for each extension number. Nonetheless, these features are manually enabled/disabled by the user, which results in the possibility of users forgetting to enable/disable these features when they leave the office or come back.

On the other hand, LTC is automatically configured for all employees and is dynamically updated in a near real-time fashion using the Message Queuing Telemetry Transport (MQTT) [9] protocol. LTC provides the highest availability of employees wherever they are within the workplace while not slowing down the organization's workflow. In addition, LTC leverages the emergence of chatbots to provide an easy way to call persons by simply stating their name or using an inquiry question to the embedded chatbot agent. This eliminates the default crucial need to know the extension number of the called person, going through complex and confusing trees to reach the right representative, or making a customer repetitively listen to "your call is important to us" while the call is forwarded to a representative who does not exist in the office. This is believed to enhance the Average Waiting Time (AWT), the time spent by an inbound call to be answered. LTC can also be expanded to greater extents, as will be discussed later in Section 6.

The rest of the paper is organized as follows. Section 2 outlines the contributions of the paper. Section 3 reviews related research in the literature and highlights their contributions and shortcomings. Section 4 delineates the building blocks of the proposed system model. Section 5 provides a feasibility study and a use case implementation of the proposed LTC system. Section 6 discusses the future work considering other application scenarios of the

proposed system that could take it to greater extents. Lastly, Section 7 concludes the paper and provides final remarks.

## 2. Contributions

The contributions of this paper are as follows:

1. We promote integration between IoT and phone technologies to build intelligent ambient-aware telephony solutions that will result in an innovative office environment, better employee productivity, and increased revenues and customer satisfaction;
2. We propose a novel telephony solution to efficiently mitigate the effect of missed business calls by increasing employees' availability regardless of their mobility within the workplace;
3. We evaluate the use of tiny and cost-effective embedded Linux platforms as suitable candidates to act as PoT gateways for homes and small-to-medium-sized business domains through a quantitative study of their capacity in gracefully processing simultaneous VoIP calls;
4. We provide a reproducible methodology to estimate the maximum number of simultaneous VoIP calls an embedded platform could gracefully handle for potential PoT applications, each of which utilizes differentiated resources of the board to achieve the designated task of the application. This will help researchers and system developers select a proper platform to satisfy the application's needs.

## 3. Related Work

To the best of our knowledge, PoT [8] is the first proposition in the literature of a general-purpose, open-source, and multi-tier IoT framework based on the integration between phones (PSTN and VoIP) and IoT technologies. Previous research has targeted specific applications and employed limited phone functionalities in third party systems as their proposed IoT systems. For example, Cirani et al. [10] proposed to enable IoT devices with a SIP-based communication protocol called "CoSIP", a constrained version of the Session Initiation Protocol (SIP) used in VoIP communication, to provide constrained smart objects with a lightweight and standardized protocol to bring them IP connectivity. Since then, other research work has been intermittently emerging to provide IoT solutions with the help of some phone functionalities. Andriopoulou et al. [11], for example, proposed a SIP-based IoT gateway for healthcare applications. Sangkong et al. [12] proposed a smart postbox system that enables delivery inspection and confirmation using VoIP technologies. Other IoT and VoIP integration applications are presented in [13–15]. Basically, these applications did not embed an IP-PBX server on the IoT gateway itself. However, they depend on utilizing a SIP client or an external IP-PBX server to provide phone functionalities to their proposed systems.

Businesses typically mitigate the impact of missed calls through voicemail. Voicemail [16] is a mature feature that comes off-the-shelf with the most unified communication (UC) solutions. Voicemail stores voice messages that the caller can optionally leave to be retrieved later by the called person. The recipient can retrieve voice messages on a phone, desktop, or email account based on the UC platform utilized. However, statistics reveal that 80% of phone calls to businesses go to voicemail, and the average voicemail response rate is less than 5% [17]. James Dupuis et al. [18] proposed to send an SMS notification to the person missing a call on their desk phone. Nurhalim et al. [19] interfaced with an Asterisk system and looked for a series of SIP messages that indicate missed calls. They then extracted the called phone number from each message, queried a database for the corresponding mobile number of the called person, and send a text message if a result was found. IBM holds a patent [20] that utilizes SMS notification to a remote phone device when a missed call occurs. The approaches mentioned above, however, are somewhat similar. They do not provide a solution to maximize availability, decrease missed call occurrences, and help reduce their corresponding drawbacks. Nevertheless, they provide means to facilitate the retrieval of the missed calls list. Organizations can do better by eliminating the

utilization of voicemails in their firms and providing better availability of their employees to mitigate the impact of missed calls in the first place. Moreover, call centers have revealed that a voicemail may be seen as a negative encounter for customers and subtracts from their satisfaction. Customers usually need instant answers to their inquiries and are not willing to wait for hours for someone to call back and help [21].

## 4. LTC Architecture

### 4.1. Overview

Figure 1 depicts the proposed LTC system. The LTC system is based on the PoT architecture proposed in [8], and it inherits its enabling technologies and modular design philosophy. LTC consists of a PoT gateway, a set of WiFi-enabled RFID door-entry nodes distributed along with the entrances of the rooms within the premise, and assistive cloud-hosted services (namely, MongoDB database and Dialogflow's chatbot agent). The door entry nodes are connected to the PoT gateway through Wi-Fi. The PoT gateway acts as an MQTT broker that relays user logs at door entry nodes (publishers) to the cloud-hosted MongoDB instance (subscriber). User logs include their RFID tag numbers, the ID of the door entry that publishes the log, and the timestamp of the log occurrence. The cloud-hosted database maps the user tracking, based on their RFID tag numbers, to the extension number(s) at their current location, depending on the ID of the door entry node that pushes the log information. Upon dialing a preconfigured hotline extension number, the call is auto answered by the embedded Asterisk server of the PoT gateway, and it invokes a developed Python script. The script saves the user's utterance as an audio file, sends it to the developed Dialogflow's chatbot agent, and waits for a reply. The chatbot agent recognizes the user's intent, identifies entities in the user speech, and determines the target employee to be called based on the exact name mentioned or the question asked by the calling person. The chatbot then sends a webhook request to the cloud-hosted database querying the extension number at the current location of the target person to be called. Upon reception of the extension number, the Dialogflow chatbot sends a text file to the PoT gateway that contains the extension number. The Python script reads the received response file from Dialogflow and extracts the extension number. The script then passes the extension number to the Asterisk server running on the PoT gateway to originate a call to that extension number and pass the call to the calling person. The following subsections delineate the system's components and highlight the motivation behind the choice of the enabling technologies they utilize.

### 4.2. PoT Gateway

#### 4.2.1. PoT Gateway as an OpenVPN Client

The PoT gateway is built upon an embedded Linux platform, namely, Raspberry Pi 4 Model B (8GB RAM) board [22]. The PoT gateway is connected to the Internet using a wired connection utilizing the built-in Fast Ethernet port on the Raspberry Pi board. The Raspberry Pi board is configured as an OpenVPN [23] client. At the startup of the board, it automatically connects through the Internet to an OpenVPN server running on a Linux-based virtual private server (VPS) hosted by Vultr [24]. The basic specifications of the VPS instance are listed in Table 1. With scalability, security, and future upgrades in mind, the encrypted VPN tunnel between the PoT gateway and the VPS instance is used to exchange all the system traffic, including VoIP and MQTT traffic. This includes the traffic between the PoT gateway and the VPS on the one hand, and the traffic between many distributed PoT gateways, as the system scales, on the other hand, as depicted in Figure 2.

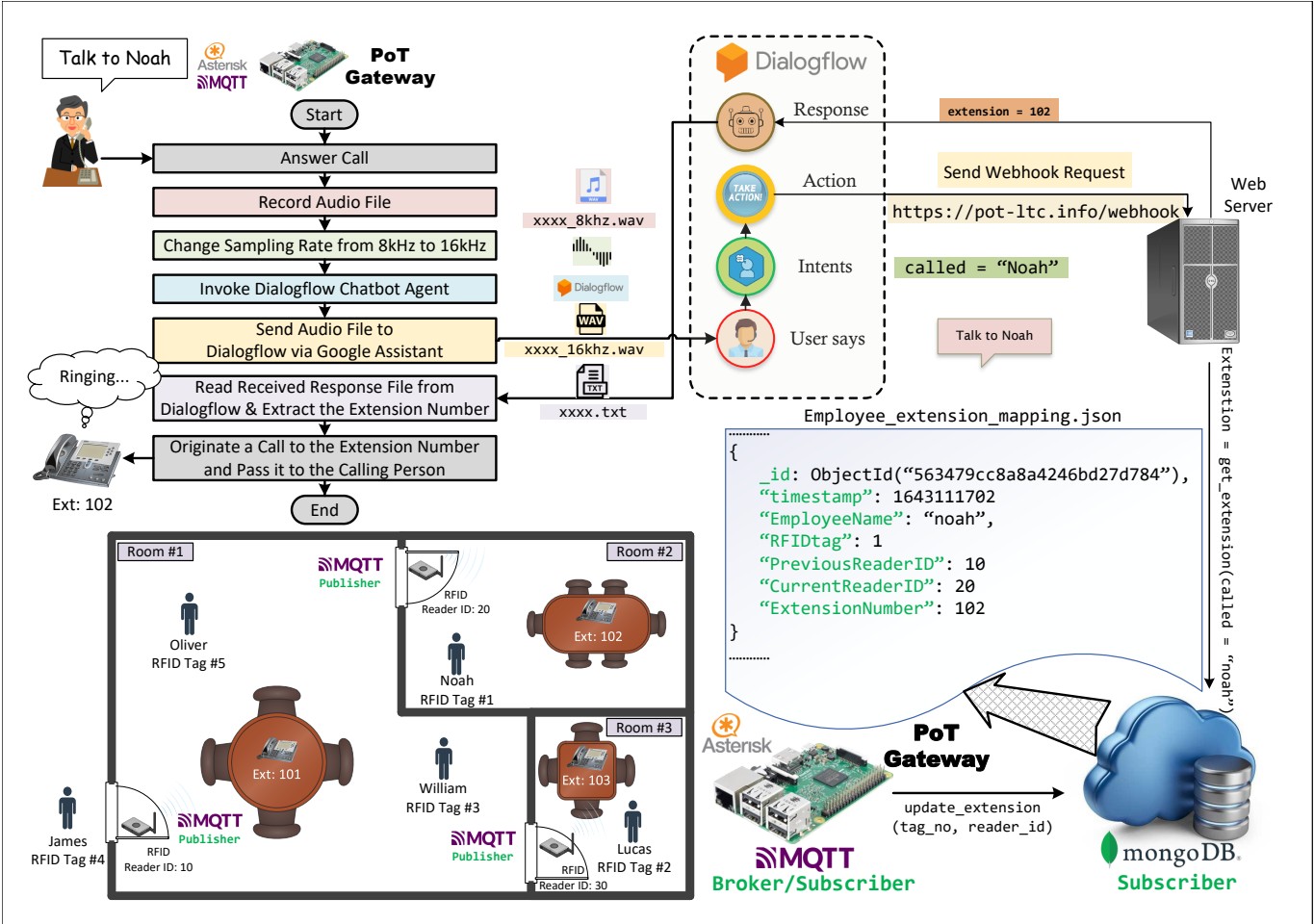

**Figure 1.** Location Transparency Call (LTC) system overview.

**Table 1.** Basic configuration of the VPS instance used in LTC.

| Property | Configuration |
|---|---|
| Server Location | Toronto, Canada |
| Server Type | Linux, Ubuntu 18.04 × 64 |
| No. of CPUs | 1 |
| RAM | 1 GB |
| Storage | 25 GB SSD |
| Bandwidth | 500 GB |
| Cost | $5 USD/month |

### 4.2.2. PoT Gateway as a Wi-Fi Access Point

The PoT gateway is configured as an access point using the built-in dual-band WLAN interface on Raspberry Pi, as depicted in Figure 3. The access point is used to connect to the distributed WiFi-enabled door entry nodes of the LTC system. It allows the PoT gateway to act as an autonomous router to handle the door entry nodes' traffic. Additionally, it provides seamless integration to the existing network infrastructure within the premise, without further configuration to handle the system traffic. The PoT gateway programmatically, dynamically, and securely handles the system's network configuration, even when the system scales up, using software-defined networking (SDN) [25] technology under the hood.

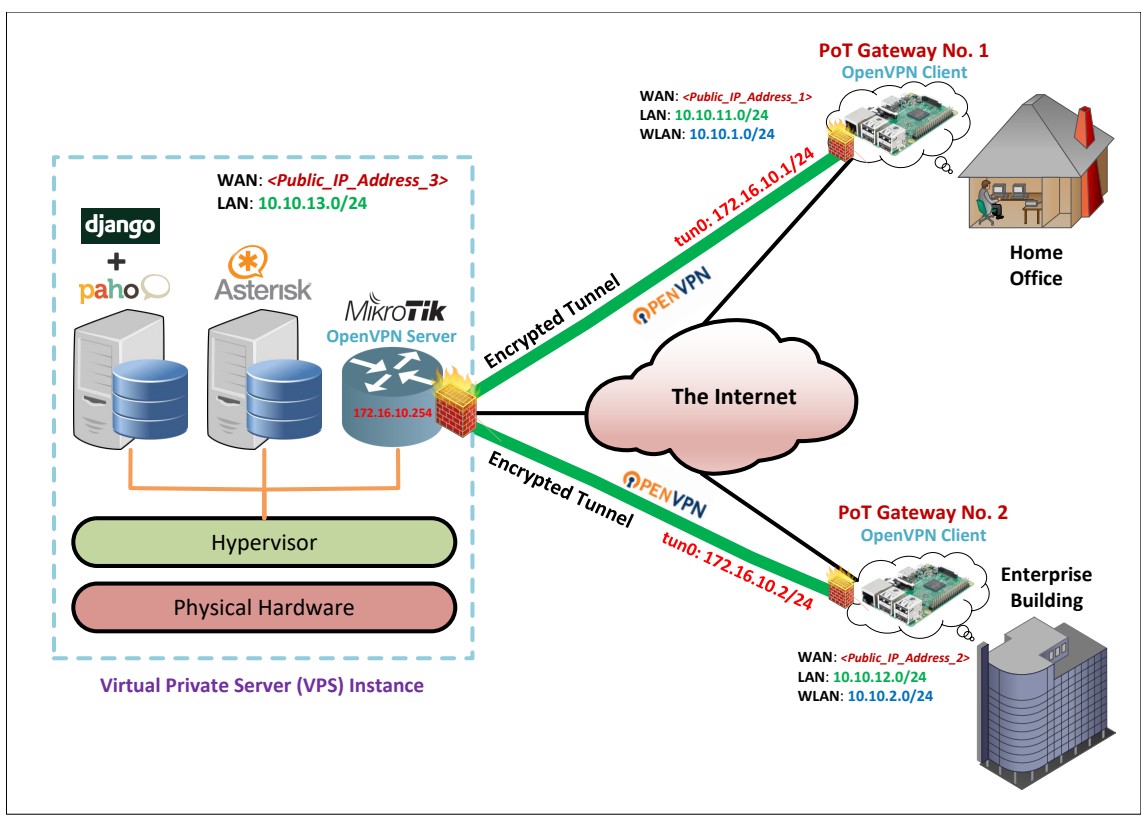

**Figure 2.** Overview of LTC OpenVPN implementation to establish secure tunnels for traffic exchange.

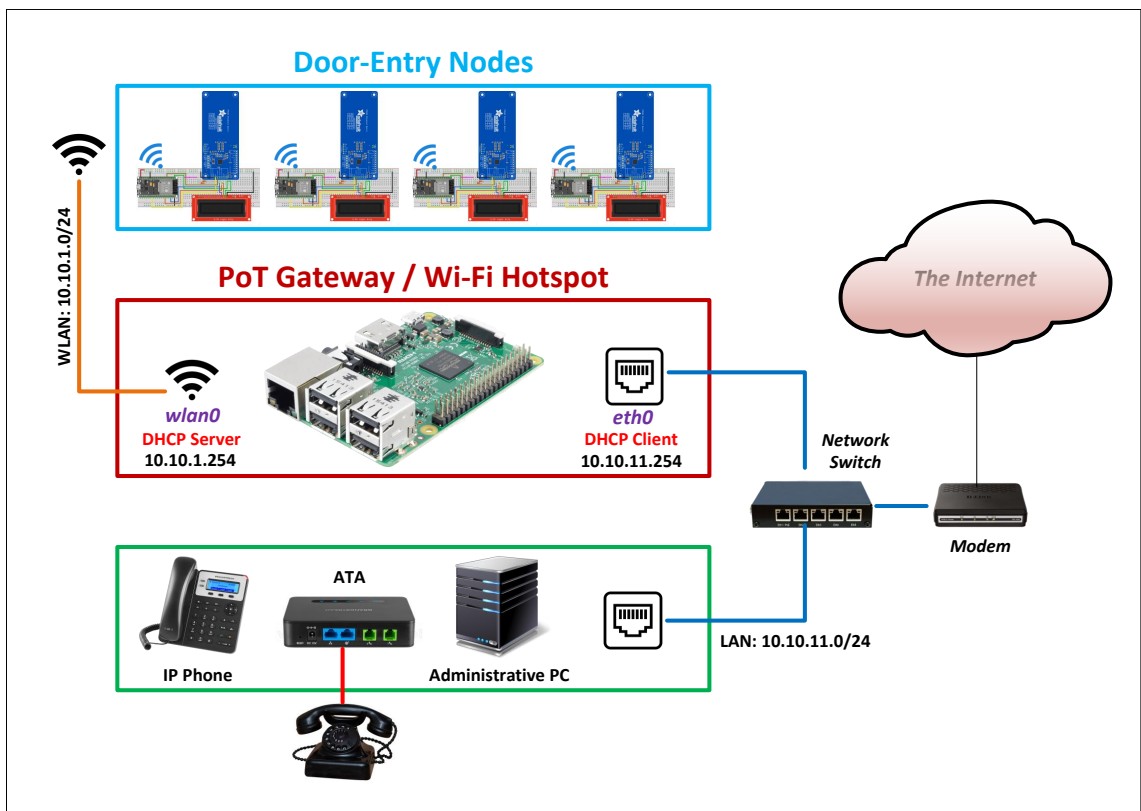

**Figure 3.** Overview of the PoT gateway as a Wi-Fi access point to the door-entry nodes.

### 4.2.3. PoT Gateway as an IP-PBX Server

The PoT gateway is equipped with an open-source IP-PBX server through the installation of a bare-metal Asterisk [26] software on top of the Raspberry Pi OS [27] (previously called Raspbian). The embedded Asterisk server to the PoT gateway can provide VoIP communication and the LTC capability to home and small business domains or as a seamless integral part of the existing UC solution within a large enterprise to enrich it with the LTC capability. In the latter case, the PoT gateway is interfaced to the existing UC through the use of phone technologies, namely, SIP (Session Initiation Protocol) trunking [28]. SIP trunking is a virtual connection that leverages the existing TCP/IP network infrastructure, ultimately the Internet, to provide for voice and video communications in UC solutions. SIP trunking eliminates the use of a dedicated physical connection between the communicating UCs, which reduces the cost and eradicates wasted resources. Additionally, since SIP trunking is a virtual connection, adding lines and modifying services are seamless and dynamic, which enhances the LTC system scalability to serve multi-site companies, as illustrated in Figure 4.

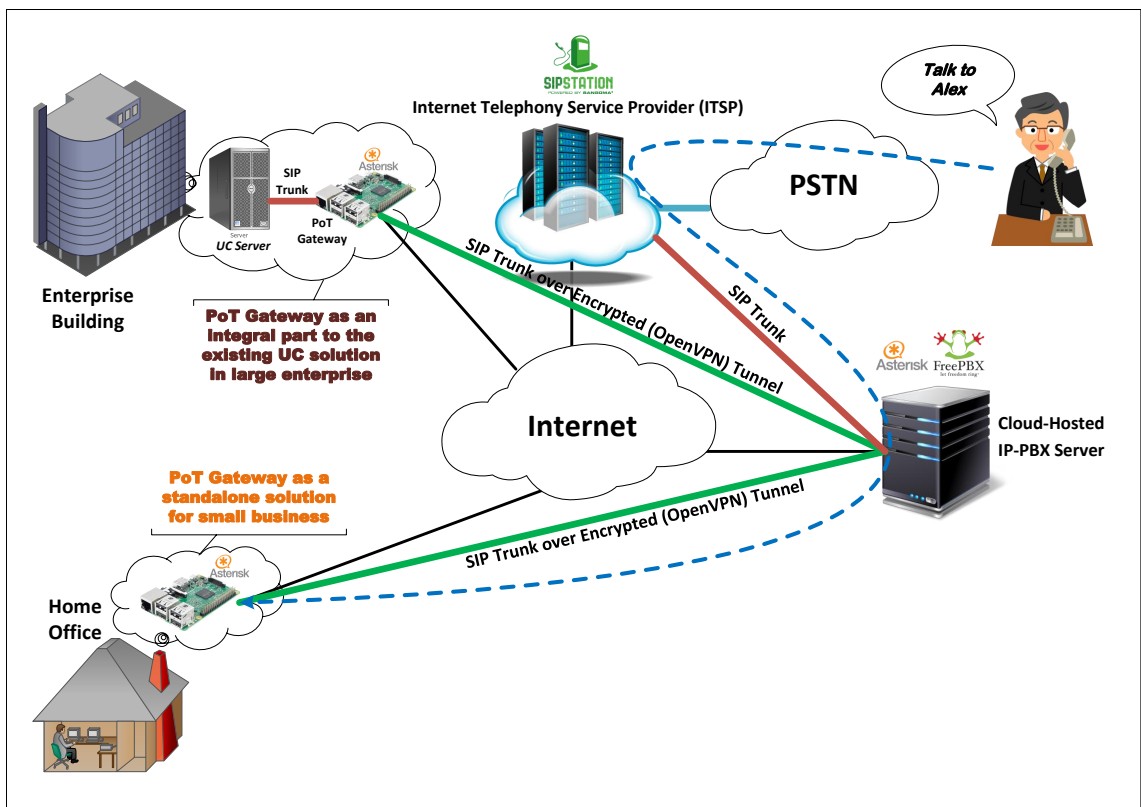

**Figure 4.** Overview of the LTC solution to serve multi-site companies.

### 4.2.4. PoT Gateway and Dialogflow Integration

With slight tweaks to the original interactional behavior between the PoT gateway and the Dialogflow [29] proposed in [8], the PoT gateway is configured such that once a user picks up the phone, the embedded Asterisk server waits for the user utterance. The embedded Asterisk server records the user's utterance as an audio file and saves it to the PoT gateway. The PoT gateway then changes the recorded audio file's sampling rate from 8 kHz (the sampling rate of phone calls) to 16 kHz (suitable for Google Assistant) and sends the file using the low-level API offered by Google to Dialogflow via the Internet and waits for a reply. The user utterance may contain the name of the called employee or a customer's inquiry to be answered. Upon the reception of the text file from Dialogflow, which includes the phone number at the current location of the target person, the system passes the number to the embedded Asterisk server to originate a call to that number. Asterisk interfacing

and audio file manipulation are carried out using Python scripts, which are invoked by Asterisk using Application Gateway Interface (AGI) [30].

### 4.2.5. PoT Gateway as an MQTT Broker

Tracking a user's location is carried out using Message Queuing Telemetry Transport (MQTT) [9]. Following the typical MQTT system architecture, the PoT gateway embeds an MQTT broker, which acts as a transaction server that orchestrates message exchange between publishers (in this case, the door entry nodes) and subscribers (a cloud-hosted web server). The web server is hosted as a virtual machine on the same VPS instance that hosts the OpenVPN server, as depicted in Figure 2. The web server is used to update an instance of MongoDB Atlas [31], a cloud Database-as-a-Service (DBaaS) hosted by MongoDB. The database is made available to Dialogflow via webhook requests to the web server, as depicted in Figure 2.

### 4.3. WiFi-Enabled RFID Door Entry Nodes

Door entry nodes are a vital component of the LTC system, and they are used to track users' current locations while they move inside the enterprise. In this work, we use the Radio Frequency Identification (RFID) [32] technology to track the users' mobility while they enter or exit the rooms. The door entry node, depicted in Figure 5, mainly consists of an RFID reader and a Wi-Fi module. The door entry node is meant to be simple, cheap, and easy to install and integrate under different circumstances without potential infrastructure modification. The proposed door entry node uses NXP's PN532 Near Field Communication (NFC) controller for RFID functionalities and Espressif's ESP32 Microcontroller Unit (MCU) for Wi-Fi connectivity. When a user carrying an RFID tag passes by a door entry node, the door entry node (MQTT publisher) reads the RFID tag and publishes the tag number and the reader ID to the MQTT broker (the PoT gateway). The web server (MQTT subscriber), which subscribes to all users' topics by default, receives the update from the PoT gateway and updates the MongoDB database accordingly. The MongoDB database is a dynamic mapping of the nearest extension number to a specific user based on his current location. The nearest extension number is determined by the current door entry checkpoint, the previous door entry checkpoint, and the preconfigured table created at the system startup. This preconfigured table maps the checkpoints' sequence to the location, based on the floor plan of the premise. Section 5 discusses the implemented algorithm to determine whether a user enters or exits a room using a single door entry checkpoint for each room.

It is worth noting that the emergence of BLE/LoRa tags and beacons also proposes suitable and cost-effective indoor positioning candidates for LTC applications. Moreover, LTC can take advantage of the wireless distribution system (WDS) in an enterprise to track people's location with no further infrastructure requirements. This is believed to reduce the system cost; since the LTC system would only need the PoT-enabled gateway to function. This also enhances the organizations' workflow; since the system would neither require the employees to tap their RFID tags every time they enter or exit a place nor need expensive long-range RFID readers to be utilized by the door entry checkpoints.

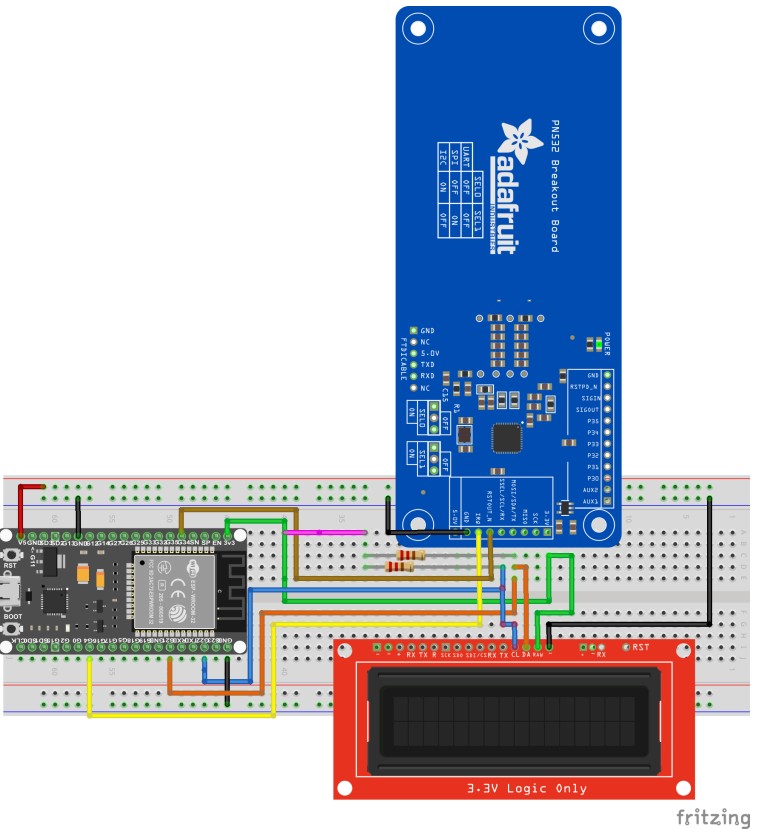

**Figure 5.** The door entry node (breadboard view).

## 5. Feasibility Study

### 5.1. Performance Evaluation of Raspberry Pi Boards as Asterisk Server for LTC Applications

5.1.1. Overview

This subsection evaluates the performance of Raspberry Pi boards that run Raspberry Pi OS (previously called Raspbian) when acting as VoIP communication servers using the open-source Asterisk 16 IP-PBX, which is installed on top of the Raspberry Pi OS. This comparative study was performed to promote the use of tiny embedded Linux platforms as suitable candidates for LTC applications in home and small business domains or as an integral part of the UC solutions in large enterprises to empower them with LTC capability.

We gauged the resource utilization and the operating system responsiveness of different Raspberry Pi boards with differentiated capabilities while initiating an increasing number of simultaneous VoIP calls. Both passthrough and transcoded VoIP calls were implemented, tested, and contrasted to give the reader a comprehensive overview of the boards' performance at different VoIP setups. We also monitored the VoIP call quality metrics, namely, jitter and packet loss, throughout each scenario to ensure it falls within the acceptable limits set by Cisco [33].

We utilized the following Python libraries in the developed Python testing scripts: (1) `pycall`: it was used for creating and using Asterisk files, and it enables programmatic origination of VoIP calls through the testing script, (2) `psutil`: it was used to retrieve information on running processes and the system performance measurements (%CPU utilization and load average), (3) `os`: it was used to run Linux bash commands from within the testing scripts and send remote commands to the peer PoT gateway used during the transcoding VoIP testing, and (4) `subprocess`: it was used to spawn new processes, connect to their input/output/error pipes, and get their return codes. The outputs of the running processes were saved within the gateway and used for further analysis and visualization using a data visualization software, namely, Tableau desktop.

### 5.1.2. Methodology

Figure 6 depicts the flowchart of the proposed performance testing algorithm:

1. A music-on-hold (MOH) extension was configured on the embedded Asterisk server of the PoT gateway. MOH maintained each originated call to that extension live until the calling party hung up the call. MOH allows one to programmatically originate an increasing number of simultaneous calls at the PoT gateway.

2. After each call origination, the operating system metrics were captured by the testing script and compared against predefined threshold values defined in the script. Thresholds were set such that %CPU utilization <75% for single-core ARM boards and <300% for quad-core ARM boards (equivalent to 75% per core) and the load average (5 min interval) < 4. These thresholds aimed at: (1) protecting the embedded platform from potential hardware damage due to the excessive CPU utilization by the Asterisk process, (2) maintaining the responsiveness of the operating system in order to be able to carry out the evaluation measurements, (3) maintaining the VoIP call quality metrics at acceptable values, and (4) determining the maximum number of simultaneous VoIP calls that the embedded platform can gracefully serve.

3. Another VoIP call was originated at the beginning of the test using an IP phone set. It called an MOH extension, and it was kept alive throughout the testing procedure. The traffic of this undergoing VoIP call was captured using Wireshark. The traffic capture was used to monitor the VoIP call quality metrics while originating an increasing number of VoIP calls during the test procedure to assure that both jitter and packet loss of the established VoIP calls still fall within the standard acceptable limits.

### 5.1.3. Passthrough VoIP Testing Overview

Figure 7 gives an overview of the passthrough VoIP testing procedure. In passthrough, the calling parties use the same codec to transform analog audio signals into a digital format suitable for transmission over the network. For passthrough performance evaluation, the testing script forces the embedded Asterisk server to originate a call to the MOH extension through the loopback interface of the board. This resembles a SIP trunk operation. The call origination by the script creates a communication channel with the Asterisk server (Channel #1). On the other hand, passing the call to the called party (MOH extension) is carried out by a SIP trunk interface defined in `pjsip_wizard.conf` configuration file that uses the loopback interface of the board (Channel #2). Since both Channel #1 and Channel #2 are configured to use the same audio codec (ulaw), a passthrough VoIP call is established between the script and the MOH extension. This configuration has two advantages: (1) It allows to perform the passthrough testing using a single Raspberry Pi board, and (2) It shortens the testing time; since each originated call actually represents two active channels. As a result, it reduces the testing time by 50%.

### 5.1.4. Transcoding VoIP Testing Overview

An overview of transcoding VoIP testing procedures is shown in Figure 8. Transcoding is the process of decoding an already encoded digital audio signal, making some alteration to its content, and encoding it again using another format. Transcoding is a process-intensive operation that consumes more resources than passthrough. In the transcoded VoIP testing, we utilized another Raspberry Pi board. The second board was equipped with a bare-metal Asterisk server, without the need for other applications or services running on the PoT gateway. The embedded Asterisk servers on the boards were interconnected using two distinct SIP trunks. The SIP trunks were configured such that they used different audio codecs. SIP trunk 1 utilized ulaw audio codec, and SIP trunk 2 utilizes g722 audio codec. We used these codecs because they are popular, open-source, and can be freely utilized by any VoIP system. The second Raspberry Pi board was used to run the test script. The test script called an MOH extension on the board (Raspberry Pi 2). However, the script forces the call to go through SIP trunk 1, which utilizes ulaw audio codec. Once the call was received at Raspberry Pi 1 (PoT gateway), it went through the extensions

defined in the `extensions.conf` configuration file on the PoT gateway to find a match. The extension definition of MOH on the PoT gateway was defined such that it went through SIP trunk 2. Since the audio codecs of the incoming channel (SIP trunk 1) and the outgoing channel (SIP trunk 2) were different, transcoding was performed on the PoT gateway prior to passing the call back to Raspberry Pi 2 through SIP trunk 2. Therefore, we could monitor the performance of the PoT gateway while carrying out transcoding operations. Since the test script was running on Raspberry Pi 2, we utilized `os` and `subprocess` Python libraries to send remote Linux bash commands from Raspberry Pi 2 to the PoT gateway. These commands were used to retrieve information about the performance measurements of the PoT gateway (%CPU utilization and load average).

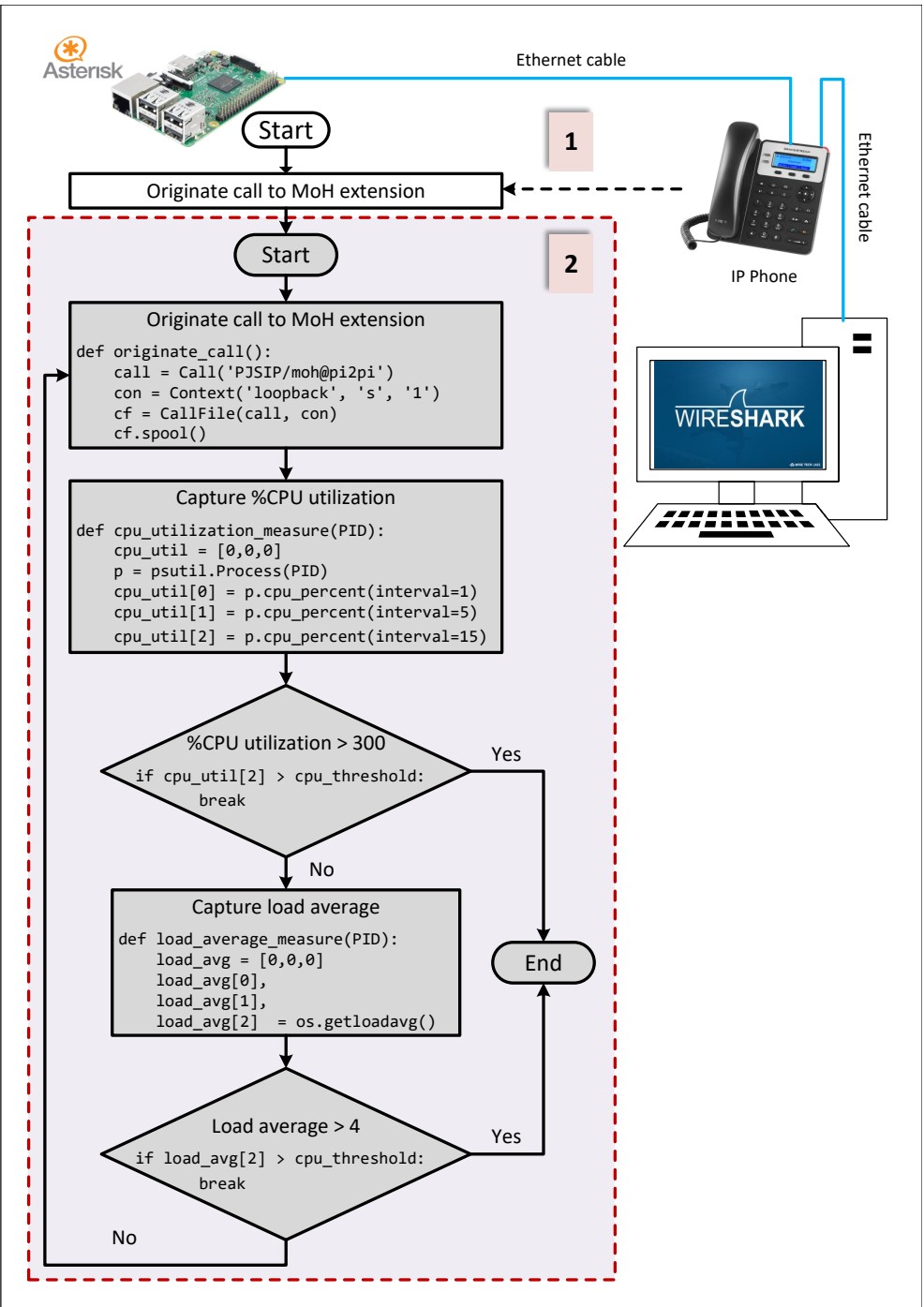

**Figure 6.** The flowchart of the proposed performance testing algorithm.

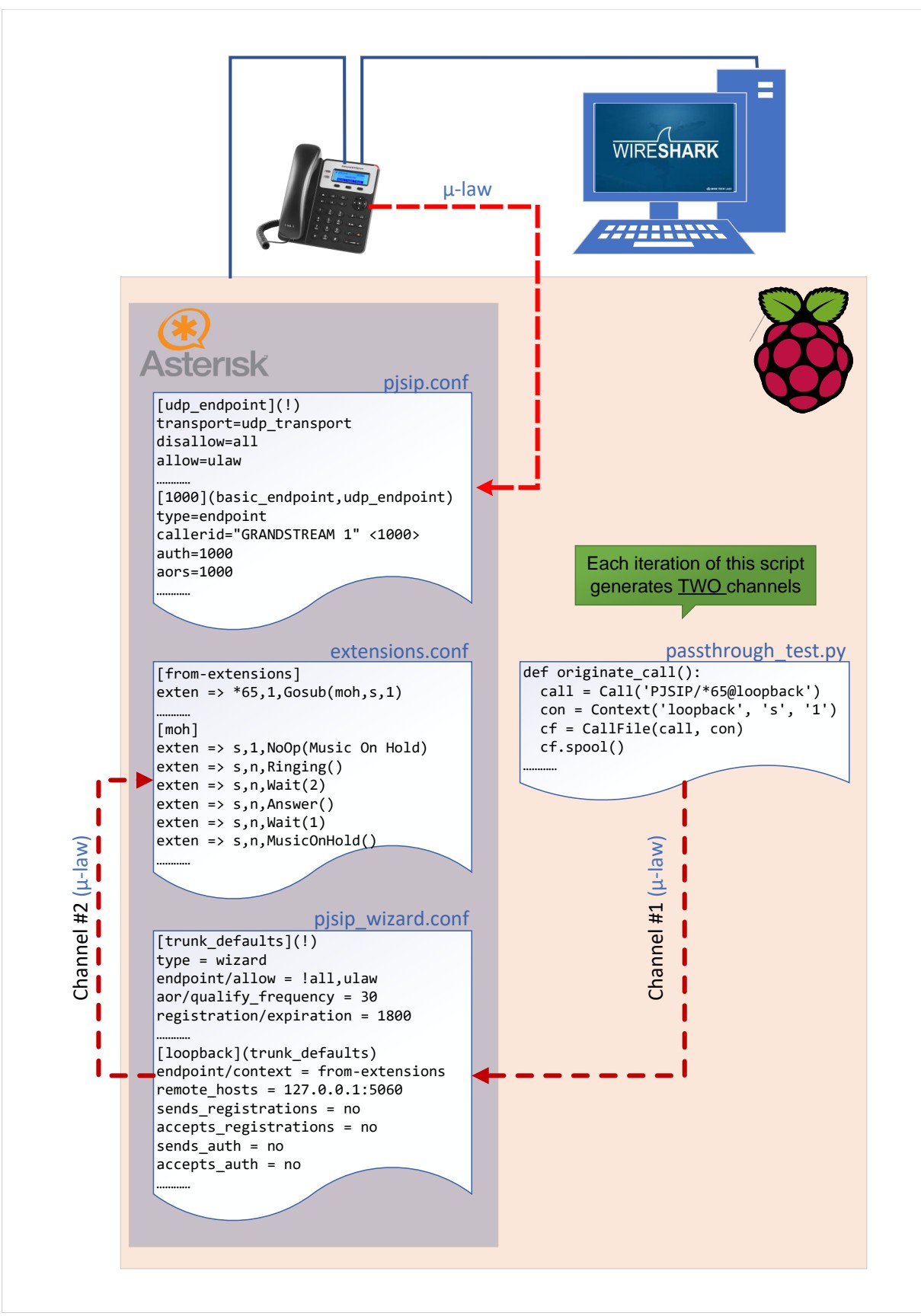

**Figure 7.** Passthrough VoIP testing overview.

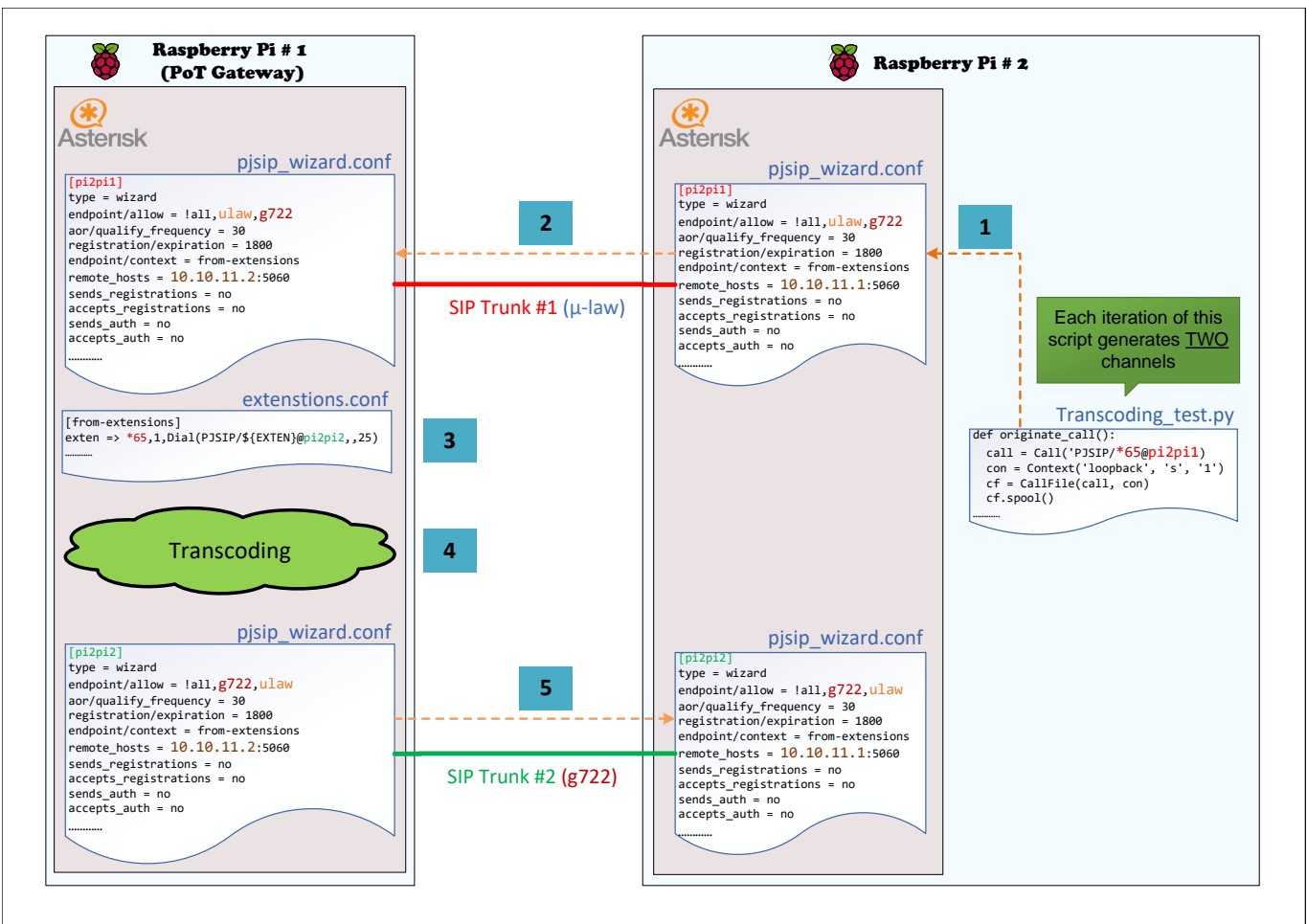

**Figure 8.** Transcoded VoIP testing overview.

5.1.5. Results

1.  Passthrough VoIP Testing

    The performance evaluation of the Raspberry Pi boards when processing passthrough VoIP calls is shown in Figures 9 and 10. The maximum numbers of simultaneous calls that the boards can gracefully handle are contrasted in Figure 11. Obviously, the maximum number of simultaneous calls is proportional to the board's hardware specifications concerning the CPU core type, number of cores, and memory size. The results show that the least powerful model of the Raspberry Pi board family, namely, Raspberry Pi Zero W, can gracefully handle up to 24 active channels, which represents 12 simultaneous calls, after which the load average starts to exceed one. The CPU utilization of the board at this maximum number of simultaneous calls is 94%. On the other hand, Raspberry Pi 4 B, the current most powerful model of the Raspberry Pi board family, can gracefully handle up to 364 active passthrough channels (182 simultaneous calls) before the five minute load average starts to exceed four. The CPU utilization of the board at this number of simultaneous passthrough VoIP calls is 266% (66.5% per core). For other Raspberry Pi models, the maximum number of simultaneous calls lie between Raspberry Pi Zero W and Raspberry Pi 4 B based on the board's hardware specifications.

    It is worth noting that Raspberry Pi Zero 2 W, the newest member of the Raspberry Pi boards family that launched in October 2021, has comparable performance to Raspberry Pi 3 B+. Raspberry Pi Zero 2 W comes at the same form factor as Raspberry Pi Zero W with a neglectable price increase ($19 CAD for Raspberry Pi Zero 2 W compared to $15 CAD for Raspberry Pi Zero W). However, Raspberry Pi Zero 2 W

comes with a quad-core ARM Cortex A53 processor, in contrast to the single-core ARM11 that comes with Raspberry Pi Zero W. The results show that Raspberry Pi Zero 2 W can gracefully serve up to 278 active channels, whereas Raspberry Pi 3 B+ can gracefully serve 308 active channels. Therefore, Raspberry Pi Zero 2 W can ideally fit PoT applications where physical size and power consumption are the primary concerns.

Except for Raspberry Pi Zero W, the performance of the Raspberry Pi boards used in the evaluation exceeded the VPS instance's performance, whose specifications are listed in Table 1. Raspberry Pi Zero 2 W, for instance, can support an approximately 70% higher number of active channels than the number of active channels supported by the VPS. It is also worth noting that the boards can still serve more VoIP calls if we switch the threshold to the 15 min load average instead of the 5 min load average or raise the threshold limit a little bit beyond four. However, when the load average exceeds four, some VoIP call processes will be queued, waiting to be served by the operating system. This queuing affects the soft real-time constraints of VoIP, and hence it degrades the resulting quality measurements of the established VoIP calls.

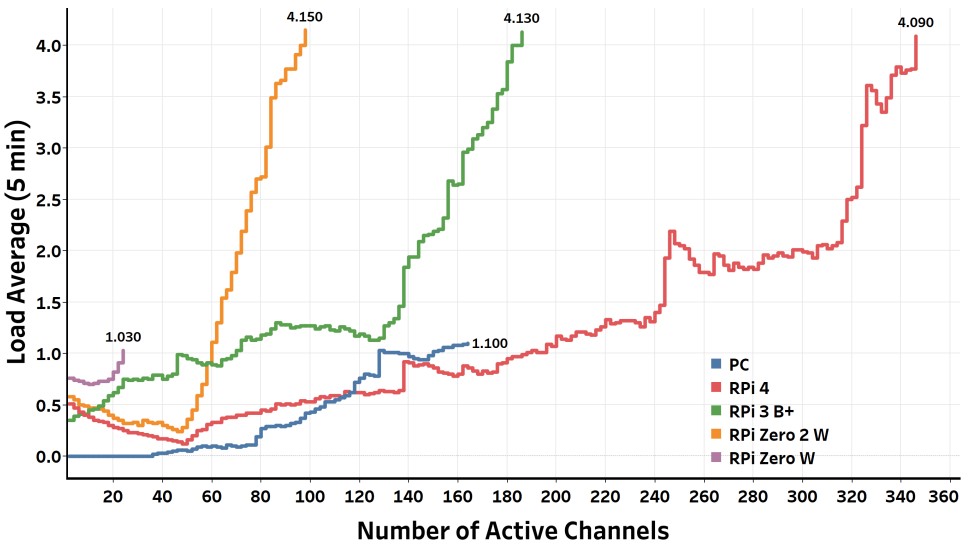

**Figure 9.** Relation between the number of active channels and the operating system load average (passthrough).

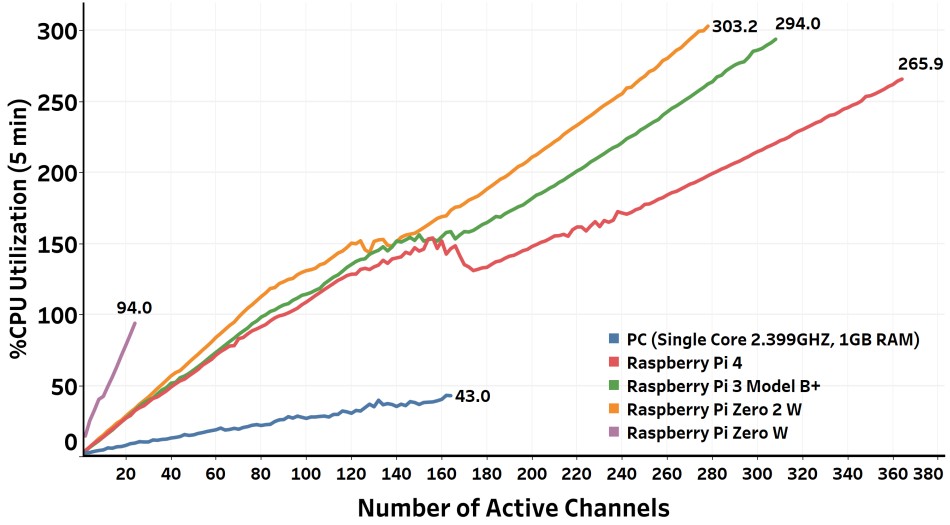

**Figure 10.** Relation between the number of active channels and the CPU utilization (passthrough).

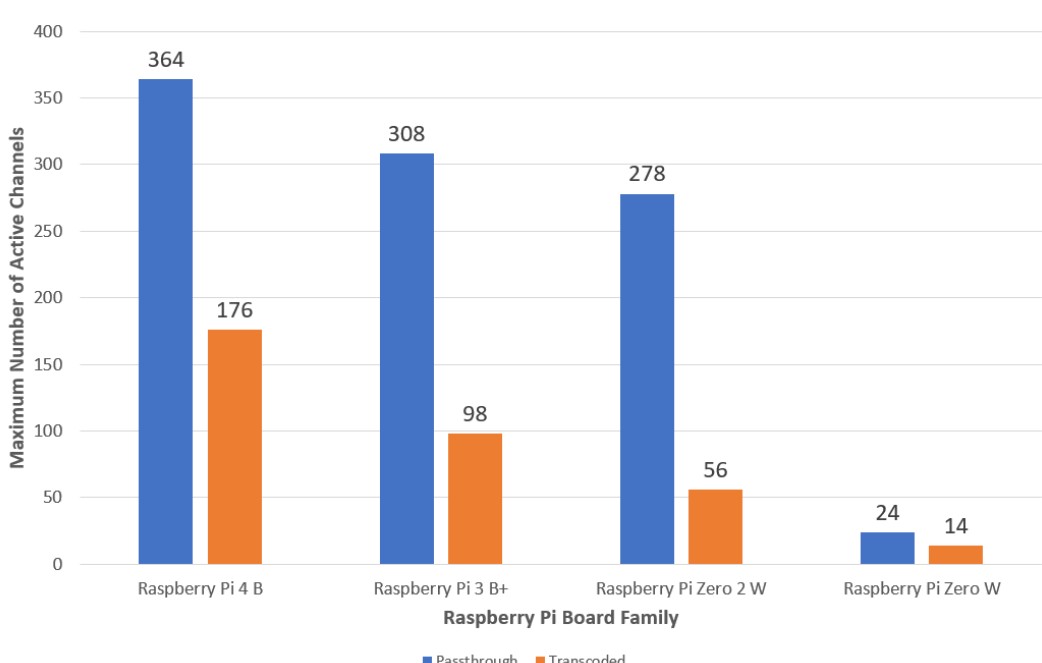

**Figure 11.** Contrasting the maximum number of simultaneous active channels that different Raspberry Pi board families can gracefully serve.

2.  Transcoded VoIP Testing

    The testing results of the simultaneous transcoded VoIP call capacity that the Raspberry Pi boards can safely process are shown in Figures 12 and 13. In contrast to passthrough VoIP, transcoded VoIP is a process-intensive task that consumes many resources. Therefore, it is obviously expected that a smaller number of simultaneous transcoded VoIP calls can be afforded than simultaneous passthrough VoIP calls using the same platform. After using the same thresholds imposed on the testing algorithm when performing the passthrough testing, the results showed that the Raspberry Pi 4 B board, for example, can gracefully handle up to 176 active channels (88 simultaneous calls) before outpacing the thresholds. This represents 48% of passthrough VoIP calls the same board can afford. The ratio between the number of graceful transcoded VoIP calls to the number of graceful passthrough VoIP calls is proportional to the processing power of the board. The higher the processing power of the board, the higher the ratio of transcoded to passthrough VoIP calls the board can gracefully afford. Concerning the VoIP call quality metrics of established calls throughout the testing procedure, the results from Wireshark, as summarized in Table 2, show values of jitter and packet loss that all fall within the acceptable standardized limits of VoIP call quality metrics.

3.  Passthrough vs. Transcoded VoIP Calls

    It is worth noting that the number of maximum simultaneous transcoded VoIP calls represents an extreme, where every originated call needs transcoding, of what actually happens in real situations. In most real cases, it is more likely that transcoding is avoided by enforcing particular audio codecs to be utilized by the communicating parties. Transcoding only happens when interconnecting with third-party providers, e.g., Internet Telephony Service Providers (ITSP), that operate other audio codecs to what we have already utilized. Therefore, we can posit that the maximum number of simultaneous VoIP calls that the Raspberry Pi boards can gracefully serve lays somewhere between the maximum number of passthrough VoIP calls and the maximum number of transcoded VoIP calls the specific platform can gracefully serve. The exact number of simultaneous VoIP calls depends on the application that defines the expected setup of the established VoIP calls.

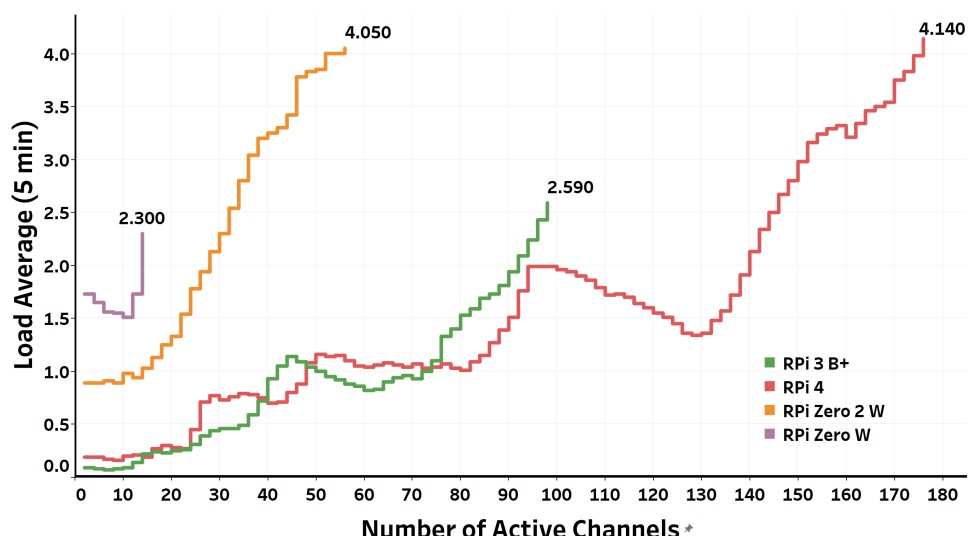

**Figure 12.** Relation between the number of active channels and the operating system load average (transcoding).

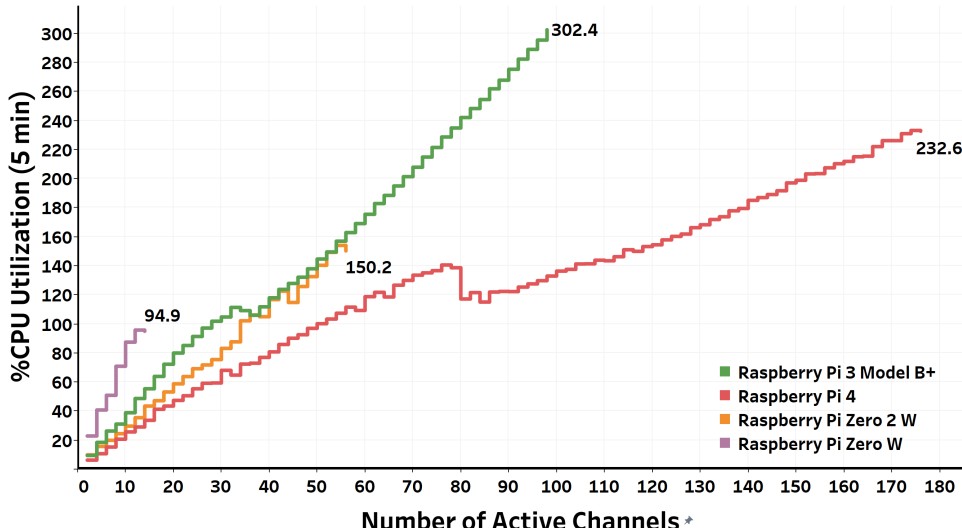

**Figure 13.** Relation between the number of active channels and the CPU utilization (transcoding).

### 5.2. LTC System for Large Enterprises

The proposed LTC system depends on an embedded Linux platform to provide VoIP functionalities along with LTC capabilities. Based on the performance evaluation carried out in Section 5.2, this meets home and small business requirements. However, this setup would not be suitable for large enterprises. As a result, for LTC deployment in large enterprises, the PoT gateway can be added as an integral part of the existing UC solution to enrich it with LTC capabilities. Otherwise, a powerful computing platform (servers, workstations, etc.) can be utilized rather than embedded Linux platforms. Figure 14 contrasts the LTC implementation for small offices and large enterprises.

**Table 2.** VoIP call quality measurements during passthrough and transcoded VoIP call testings for different Raspberry Pi boards.

| | RPi 4 B | | RPi 3 B+ | | RPi Z 2 W | | RPi Z W | |
|---|---|---|---|---|---|---|---|---|
| | Passthrough | Trans. | Passthrough | Trans. | Passthrough | Trans. | Passthrough | Trans. |
| **Forward (Phone-to-RPi)** | | | | | | | | |
| Max jitter (milliseconds) | 9.35 | 10.49 | 11.87 | 10.97 | 12.05 | 13.85 | 15.33 | 16.02 |
| Mean jitter (milliseconds) | 7.08 | 6.45 | 8.09 | 6.88 | 9.96 | 10.13 | 11.58 | 13.69 |
| RTP Packets | 132,486 | 115,715 | 111,460 | 65,414 | 101,623 | 37,211 | 8688 | 5058 |
| Expected | 132,486 | 115,715 | 111,460 | 65,414 | 101,623 | 37,211 | 8688 | 5058 |
| Packet loss (%) | 0.0 | 0.0 | 0.0 | 0.0 | 0.0 | 0.0 | 0.0 | 0.0 |
| **Reverse (RPi-to-Phone)** | | | | | | | | |
| Max jitter (milliseconds) | 5.07 | 5.98 | 5.90 | 8.43 | 8.85 | 9.32 | 10.02 | 20.36 |
| Mean jitter (milliseconds) | 0.65 | 0.68 | 0.70 | 0.73 | 0.98 | 0.85 | 0.90 | 3.5 |
| RTP Packets | 132,450 | 115,624 | 111,365 | 65,369 | 101,577 | 37,188 | 8610 | 5049 |
| Expected | 132,450 | 115,624 | 111,365 | 65,369 | 101,577 | 37,188 | 8610 | 5049 |
| Packet loss (%) | 0.0 | 0.0 | 0.0 | 0.0 | 0.0 | 0.0 | 0.0 | 0.0 |

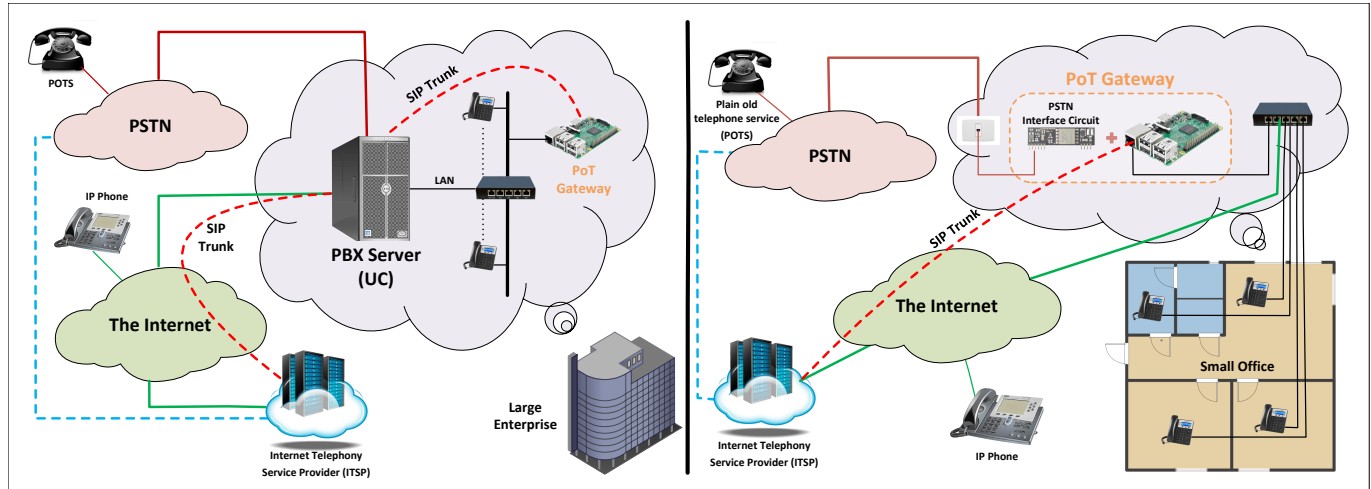

**Figure 14.** LTC implementation scenarios: a small office and a large enterprise.

### 5.3. Entering and Exiting a Place

The proposed LTC system uses a single door entry checkpoint for each room to track the users' mobility. The architecture of the door entry node, depicted in Figure 5, depends on the RFID reader, and it does not utilize any other kind of sensors to help identify whether the user is entering or exiting the room. This is meant to simplify the design and the installation of the door entry node. We designed the tracking database's schema to maintain fields that hold the identities of the current door entry checkpoint and the previous door entry checkpoints. Knowing these two identities of door entry checkpoints and the truth table that is manually configured during system startup based on the floor plan of the premise, we can determine whether the user is entering or exiting a place. Moreover, we set up a fallback extension for invalid unmapped entries in the truth table. Invalid entries may be due to system glitches or missed RFID tag readings when the users enter or exit places. The fallback extension number can be configured as the default extension number of the user or as an arbitrary extension number (i.e., the receptionist extension). Figure 15

gives an example scenario that illustrates the leverage of a manually pre-built truth table and a single RFID checkpoint at each door entry to determine the user location. Table 3 shows the truth table of the example scenario. The table is a logical mapping for the specific floor plan given in the example. It states the location based on the previous (t − 1) and the current (t) identities of the door entry checkpoints. Upon system startup, all users' (t − 1) entries are initialized to void/empty. This indicates that the user does not exist within the company's premises. This methodology is easy to implement, and it does not propagate errors in case of missed or invalid reading sequences. Table 4 gives examples of valid and invalid entries based on the truth table mentioned above.

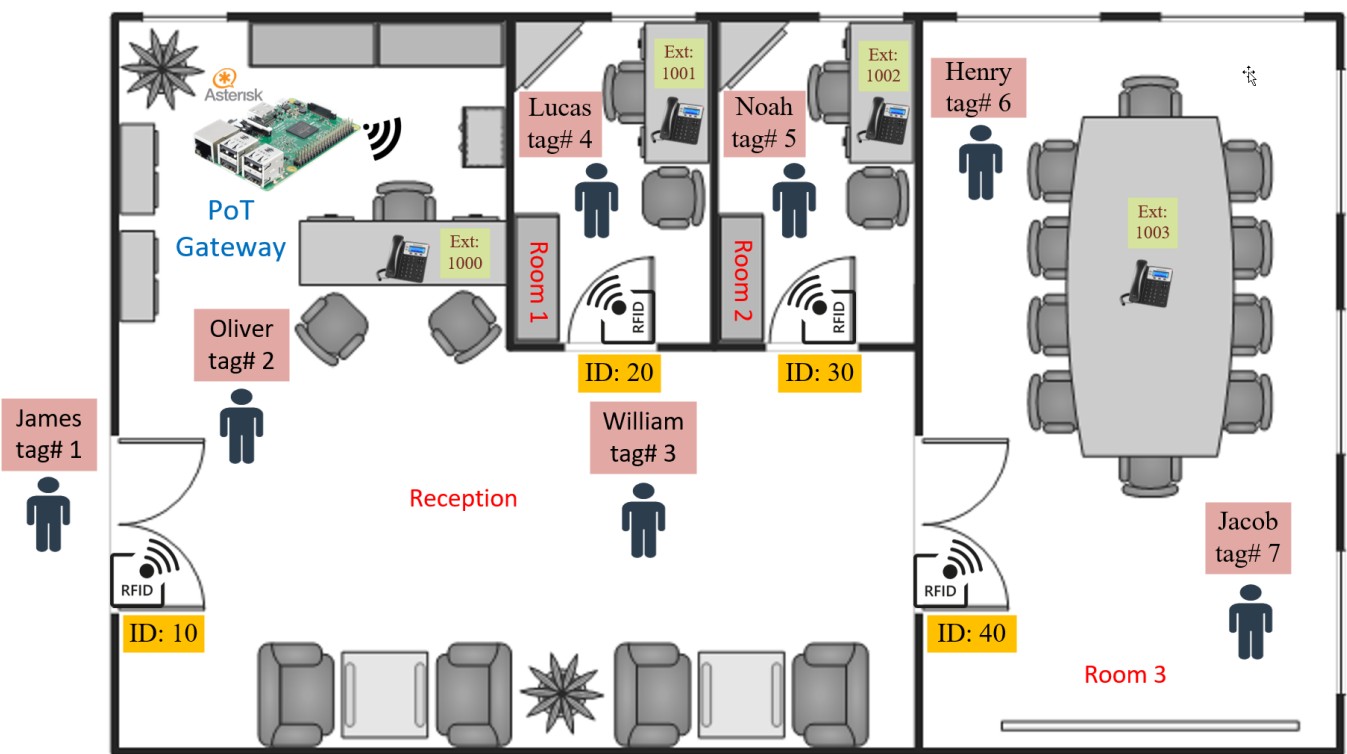

**Figure 15.** Entering and exiting example scenario.

**Table 3.** Truth table of the example scenario's floor plan.

| Reader ID (t − 1) | Reader ID (t) | Location | Extension No. | Remark |
|---|---|---|---|---|
| - | 10 | Reception | 1000 | Coming |
| 10 | 10 | Reception | 1000 | Leaving (Fallback) |
| 10 | 20 | Room 1 | 1001 | Entering Room 1 |
| 10 | 30 | Room 2 | 1002 | Entering Room 2 |
| 10 | 40 | Room 3 | 1003 | Entering Room 3 |
| 20 | 10 | Reception | 1000 | Leaving (Fallback) |
| 20 | 20 | Reception | 1000 | Entering Reception |
| 20 | 30 | Room 2 | 1002 | Entering Room 2 |
| 20 | 40 | Room 3 | 1003 | Entering Room 3 |
| 30 | 10 | Reception | 1000 | Leaving (Fallback) |
| 30 | 20 | Room 1 | 1001 | Entering Room 1 |
| 30 | 30 | Reception | 1000 | Entering Reception |
| 30 | 40 | Room 3 | 1003 | Entering Room 3 |
| 40 | 10 | Reception | 1000 | Leaving (Fallback) |
| 40 | 20 | Room 1 | 1001 | Entering Room 1 |
| 40 | 30 | Room 2 | 1002 | Entering Room 2 |
| 40 | 40 | Reception | 1000 | Entering Reception |

**Table 4.** Entering and exiting illustration example.

| No. | Name | RFID Tag No. | Reader ID (t − 1) | Reader ID (t) | Location | Extension No. |
| --- | --- | --- | --- | --- | --- | --- |
| 1 | James | 1 | 10 | 10 | Not Exist | 1000 (Fallback) |
| 2 | Oliver | 2 | - | 10 | Reception | 1000 |
| 3 | William | 3 | 20 | 20 | Reception | 1000 |
| 4 | Lucas | 4 | 10 | 20 | Room 1 | 1001 |
| 5 | Noah | 5 | 20 | 30 | Room 2 | 1002 |
| 6 | Henry | 6 | 10 | 40 | Room 3 | 1003 |
| 7 | Jacob | 7 | 30 | 40 | Room 3 | 1003 |
| 8 | Lucas | 4 | - | 20 | Invalid | 1000 (Fallback) |

*5.4. Multi-Site Support*

The LTC system can support multi-site enterprises by connecting the embedded Asterisk servers of multiple PoT gateways using virtual SIP trunks, as depicted in Figure 4. Moreover, by connecting the PoT gateway to an Internet Telephony Service Provider (ITSP) (SIPStation, Flowroute, etc.), as depicted in Figure 4, the LTC system can even be expanded to a greater extent. Even if the called person does not exist within any of the enterprise's premises, the embedded Asterisk server can automatically seize the SIP trunk from the ITSP to make an outbound call to reach the registered mobile phone of the target person.

**6. Future Work**

The proposed LTC system still has a lot to offer. Artificial intelligence (AI) can enhance the applications of LTC by choosing the optimum way to reach the called person based on some criteria. The system might prefer to call users using the internal extension at their current locations rather than their mobile phones. This can be automatically triggered if the mobile signal strength is weak, thereby reducing the mobile radiation exposure to the called persons and saving their mobiles' battery lives. Additionally, LTC can be used to build ambient-aware telephony solutions. For example, the system would automatically forward calls received at fixed home phones to the users' mobile phones if the PIR sensors are deactivated.

The LTC system still has a lot to offer. LTC can be implemented and offered as a cloud service to subscribers. Hotels and businesses would be potential beneficiaries of the cloud-hosted LTC service to enhance the engagement of their guests or increase the productivity of their employees, respectively. In this scenario, a proprietary mobile application is used that connects to the cloud-hosted LTC. All calls coming to the users' mobile phones are then automatically forwarded by the application to the extension numbers at their locations. Therefore, guests in hotels, for example, can receive their mobile calls at the extensions of their suites during their stays.

**7. Conclusions**

This paper presents the Location Transparency Call (LTC) system, which bridges IoT with phone technologies to build an intelligent telephony solution for business enterprises. LTC is an application of the Phone of Things (PoT), and it inherits its architecture. LTC provides an intelligent phone dialing solution to mitigate missed business calls. LTC tracks the mobility of the users within the premises using RFID tags carried by the users and a set of RFID-enabled nodes installed at each door entry and connected wirelessly to the PoT gateway, which maps the users to the closest extension numbers at their current locations. The PoT gateway automatically and dynamically forwards all incoming calls indented to an employee to the nearest extension number. We demonstrated the proposed system architecture and highlighted the motivation behind utilizing the enabling technologies operated by the system. We conducted a feasibility study of the proposed system by evaluating the performance of tiny and cost-effective embedded Linux platforms when acting as Asterisk servers for LTC applications. The results of the feasibility study are impressive. The results promote the utilization of embedded Linux platforms as appro-

priate candidates to act as PoT gateway in small-to-medium-sized business domains. The results show, for example, that Raspberry Pi 4 B can gracefully handle 182 simultaneous passthrough VoIP calls. However, Raspberry Pi Zero W, the least influential member of the Raspberry Pi family, can gracefully handle 12 simultaneous passthrough VoIP calls. For large-scale business applications, however, we discussed the utilization of the PoT gateway as an integral addition to the existing UC solution which will enrich it with LTC capabilities. We also discussed other possible applications of LTC to bring it expand it utility.

**Author Contributions:** Writing—original draft, H.K.; Writing—review & editing, K.E. All authors have read and agreed to the published version of the manuscript.

**Funding:** This research was funded by Natural Sciences and Engineering Research Council of Canada (NSERC) grant number CRC-2017-00170.

**Data Availability Statement:** Not Applicable, the study does not report any data.

**Conflicts of Interest:** The authors declare no conflict of interest.

## Abbreviations

The following abbreviations are used in this manuscript:

| | |
|---|---|
| AI | Artificial Intelligence |
| AGI | Application Gateway Interface |
| API | Application Programming Interface |
| ATA | Analog Telephone Adapter |
| CoSIP | Constrained Session Initiation Protocol |
| CPU | Central Processing Unit |
| DBaaS | Database as a Service |
| DIY | Do-It-Yourself |
| IoT | Internet of Things |
| IP | Internet Protocol |
| ITSP | Internet Telephony Service Provider |
| IVR | Interactive Voice Response |
| LAN | Local Area Network |
| MCU | Microcontroller Unit |
| MQTT | Message Queuing Telemetry Transport |
| NFC | Near Field Communication |
| PBX | Private Branch Exchange |
| PoT | Phone of Things |
| PSTN | Public Switched Telephone Network |
| RFID | Radio Frequency Identification |
| RTP | Real Time Protocol |
| SDN | Software-Defined Networking |
| SIP | Session Initiation Protocol |
| SMS | Short Message Service |
| UC | Unified Communication |
| VoIP | Voice over Internet Protocol |
| VPN | Virtual Private Network |
| VPS | Virtual Private Server |
| WAN | Wide Area Network |
| WDS | Wireless Distribution System |
| WLAN | Wireless Local Area Network |

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
