# Peer review of "Location Transparency Call (LTC) System: An Intelligent Phone Dialing System Based on the Phone of Things (PoT) Architecture"

_futureinternet, doi:10.3390/fi14040111_

Round 1

Reviewer 1 Report

This manuscript describes Location Transparency Call (LTC) System based on PoT architecture and is an applied scientific research work. 

The text is well-written and well-structured. The authors present in detail all the components of their PoT architecture and provide meaningful results.

My only suggestion for further work is related to RFID NFC technology limitations and the idea to test other IoT systems for indoors personnel tracking (LoRa End nodes).

some typos:

line 238 Fig. please define

line 254 ... as ...

Reviewer 2 Report

The presented paper discusses perspective approach of intelligent phone dialing system based on the phone of things architecture. Presented results looks very interesting and practical, its application can really enhance efficiency of business processes relating with dialing system applications. The main novelty of paper is in presented LTC system that can forward calls to the intended person at their current locations. Basically I do not know any person who will be satisfied of call servicing of any company, so presented in the paper results possibly can change this situation. Paper requires minor revision before it can be published, main comments are:

  1. Figure 1, 7 and 8 – figures should be a little bit bigger because some text is difficult to read;
  2. How presented in the paper architecture can influence on the time-of-wait for person who call in the company?
  3. Conclusion should be rewritten to clarify main results of the paper;
  4. lines 376-381 – it looks more like discussion that conclusion, thus it can be expanded for separate discussion section. In general discussion section can significantly improve paper – authors can discuss advantages and possible application of the presented results;
  5. Abbreviations list is necessary.

Reviewer 3 Report

This is a very interesting manuscript pertinent to the paradigm of the Phone of Things (PoT). Nevertheless, it's not appropriate by any means to label PoT as ... 'bringing a complete paradigm shift to the IoT System Design'. In fact, the PoT (as depicted in Figure 1) relies on the underlying principles of IoT. Also, I would suggest referring to the notion of IoT and then systematically linking it with the PoT. The authors thus may read the following scholarly work:

  1. The 10 Research Topics in the Internet of Things. 2020 IEEE International Conference on Collaboration and Internet Computing (CIC), pp. 34-43, doi: 10.1109/CIC50333.2020.00015.
  2. An Internet of Things Service Roadmap. Communications of the ACM, 64, 9, 86–95. doi :https://doi.org/10.1145/3464960.

Also, it is indispensable to illustrate 'Figure 1 - Location Transparency Call (LTC) System Overview' in a bit more detail.

Overall, the proposed System is quite basic in nature and I per se don't see any considerable novelty in the same. It is thus important that the contributions of the manuscript should be categorically and sequentially delineated towards the end of Section I.

Figures 6, 7, and 8 should be delineated in a bit of a more categorical manner. Furthermore, critical analysis of Figures 9 -  13 is indispensable too.

Round 2

Reviewer 3 Report

Thank you for submitting the revised version of this Manuscript. However, after looking at the Revised Version and the Response Letter (wherein, the emphasis has just remained on redirecting here and there and referring to Section 5), I am not satisfied with this Revision by any means.

Please note that the Novelty of this work is already pretty average and also somehow tends to be ambiguous for the readers. Therefore, please narrate your contributions towards the end of Section 1 in a sequential manner, i.e., as either (1) ..., (2) ..., (3) ... or as (a) ..., (b) ..., (c)... - some authors even come up with a separate Subsection, Contributions, in the Introduction. This is a structural problem and you should address it.

Re. Figure 1 - Location Transparency Call (LTC) System Overview, I am aware that the complete Architecture has been elaborated in Section 3. However, since you have done the courtesy to put it on Page 2 here, there is a need to either provide a brief key description pertinent to the same, or at the least, put a comment and refer the readers towards Section 3 for further details. Once again, this is a structural problem and you should address it.

Re. Figures 6, 7, and 8 and Figures 9 - 13, I am again very well aware that the respective description has been documented in the various Subsections of Section 4. However, you have merely provided a basic illustration here. So, a more detailed illustration is required for Figures 6, 7, and 8 that delineate the underlying process, i.e., a couple of lines per Figure would do the work here. Similarly, critical analysis is required for Figures 9 - 13, i.e., what you are illustrating can be observed by any reader by just looking at the Figures, however, you need to communicate beyond that. Also, please revisit your results and try not to use the word 'may' as it appears that you are really not sure that what is really happening in the same and are making wild guesses.

Please note that all these comments are furnished to enhance the readability of this particular manuscript and I would appreciate if they are followed in the true spirits.

Round 3

Reviewer 3 Report

Thank you for addressing the comments in a decent manner.